# A Causal World Model Underlying Next Token Prediction: Exploring GPT in a Controlled Environment

Raanan Y. Rohekar [* 1]   Yaniv Gurwicz [* 1]   Sungduk Yu [1]   Estelle Aflalo [1]   Vasudev Lal [1]

## Abstract

Are generative pre-trained transformer (GPT) models, trained only to predict the next token, implicitly learning a world model from which sequences are generated one token at a time? We address this question by deriving a causal interpretation of the attention mechanism in GPT and presenting a causal world model that arises from this interpretation. Furthermore, we propose that GPT models, at inference time, can be utilized for zero-shot causal structure learning for input sequences, and introduce a corresponding confidence score. Empirical tests were conducted in controlled environments using the setups of the Othello and Chess strategy games. A GPT, pre-trained on real-world games played with the intention of winning, was tested on out-of-distribution synthetic data consisting of sequences of random legal moves. We find that the GPT model is likely to generate legal next moves for out-of-distribution sequences for which a causal structure is encoded in the attention mechanism with high confidence. In cases where it generates illegal moves, it also fails to capture a causal structure.

## 1. Introduction

In recent years, the generative pre-trained transformer (GPT) model (Radford et al., 2018) has demonstrated high-quality generative capabilities, as perceived by humans. Although this model is trained to generate one token at a time, it has been demonstrated to perform a range of tasks beyond next-token prediction, such as visual understanding and symbolic reasoning (Liu et al., 2024; Team et al., 2023; Chowdhery et al., 2023). Are these emergent abilities (Li et al., 2023) or are they merely a 'mirage' resulting from the choice of metric and task (Schaeffer et al., 2024)?

[*]Equal contribution [1]Intel Labs. Correspondence to: Raanan Rohekar <raanan.yehezkel@intel.com>.

*Proceedings of the 42nd International Conference on Machine Learning*, Vancouver, Canada. PMLR 267, 2025. Copyright 2025 by the author(s).

In this paper, we suggest that there is no restriction in the GPT architecture that prevents it from learning conditional independence (CI) relations between tokens in a sequence. Moreover, under certain assumptions, a causal structure is directly entailed from these CI relations. One may ask whether this lack of restriction results in implicitly learning a causal model of the world during the pre-training procedure of GPT. Assuming that both a causal world model and a model based on surface statistics are sufficient solutions, one possibility is that a causal world model is more compact and more likely to be learned during pre-training, in line with Occam's razor. For example, if weights are distributed from a uniform distribution in the surface statistics model, then a causal structure limits the range of their distribution. If so, what assumptions underlie this causal world model?

Rohekar et al. (2024) recently proposed ABCD, a method for causal interpretation of unmasked self-attention in BERT models (Devlin et al., 2019), demonstrating its use in explaining movie recommendations (Nisimov et al., 2022). We take a similar approach, with key differences, and propose a causal interpretation of GPT's masked attention mechanism. Furthermore, we define a corresponding causal world model. ABCD is adapted to learn causal structures, where the induced dependency relations are encoded in GPT's attention matrices. We then ask whether errors generated by GPT are correlated with the uncertainty in representing the causal structure by the attention matrices. To this end, we define a metric based on the entropy of $p$-values from CI tests used for inferring the causal structures.

## 2. Related Work

Recent work has examined the internal process of large language models and investigated whether a world model is implicitly learned using a well-defined and constrained setting, such as in Chess (Toshniwal et al., 2022) and Othello (Li et al., 2023) games environments. For the Othello board game setting, Li et al. (2023) demonstrated that the board state can be inferred from attention matrices in GPT, and Nanda et al. (2023) showed that a linear classifier suffices to reconstruct the board state from these attention matrices. They claim the emergence of a world model in GPT. Nevertheless, they do not explain how the board game is

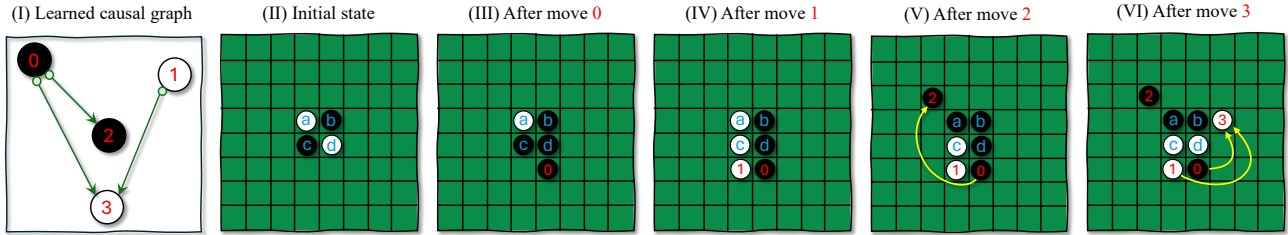

*Figure 1.* An example of a real Othello game sequence and the corresponding causal structure recovered using the proposed method. Red numbering {0,1,2,3} on the causal graph nodes and game board discs corresponds to the indices of the game moves. The blueish letters {a,b,c,d} indicate the discs in the initial state of the board game. (I) The causal graph learned by our method given the sequence of moves described hereafter. (II) The initial state of the board. (III) After move 0: Black plays and flips disc 'd' to black. (IV) After move 1: White plays and flips disc 'c' to white. This move does not depend on the previous move 0, and it aligns with the learned causal graph where node '1' is found independent of node '0'. (V) After move 2: Black plays and flips disc 'a' to black. This was made possible since disc 'd' had being flipped to black in the earlier move 0 (this causal link is indicated by a yellow arrow). Correspondingly, this causal link is also revealed in the learned causal graph by node '0' being the sole parent of node '2'. (VI) After move 3: White plays and flips disc 'd' to white. This was made possible because disc '1' was white (due to move 1), and disc 'd' was black (as mentioned before, it was flipped to black earlier in move 0). Therefore we expect both moves 0 and 1 to be the causes of move 3 (as indicated by yellow arrows). This is exactly revealed by the learned causal graph.

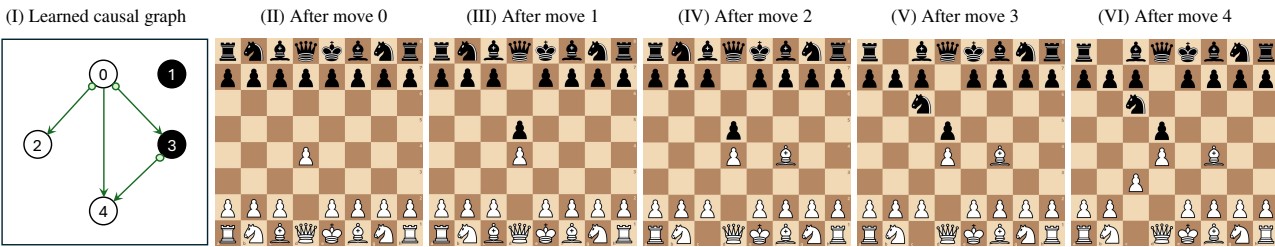

*Figure 2.* An example of a Chess game sequence and the corresponding casual structure recovered using the proposed method. It is evident that first move (Move 0), played by White, enables playing Move 2 (a directed edge from node 0 to node 2). In addition playing Move 0 led Black to play Move 3 (a directed edge from node 0 to node 3). These moves led to Move 4 (directed edges from nodes 0 and 3 into 4.

encoded within the attention matrices, nor why the attention mechanism can represent the board state. In essence, they do not provide an explanation for the apparent emergence of the world model. Furthermore, their reconstructed world model (the board game state) applies only to the domain for which the GPT model was trained and lacks the generative mechanism underlying the token sequences.

In this paper, we consider the structural causal model as a general-purpose world model that describes the generative process that is applicable across various domains (not specific to a single task, such as the board state in Othello or Chess). We explore whether GPT is capable of capturing properties of this world model, which may help explain its apparent emergence. See an example for Othello in Figure 1 and for Chess in Figure 2.

## 3. Preliminaries

In this section, we provide the notations and descriptions for self-attention in the GPT architecture, as well as for struc-

tural causal models. Matrices are written in bold, vectors in bold-italic, and models in calligraphic font. A summary of the main symbols used in this paper is provided in Table 1.

### 3.1. Attention in GPT

Attention is a mechanism that estimates network weights with respect to the context in an input sequence of tokens (Schmidhuber, 1992). In a GPT model, which is based on the decoder part of the Transformer architecture (Vaswani et al., 2017), an attention layer estimates an $n \times n$ lower-triangular (masked) attention matrix $\mathbf{A}$ given an input sequence of $n$ tokens. The input sequence is in the form of an $n \times d$ matrix $\mathbf{Y}$, where the $i$-th row vector $\mathbf{Y}(i, \cdot)$ is an embedding (representation) of the $i$-th token in $d$ dimensions. The attention matrix is estimated by $\mathbf{A} = softmax(\mathbf{Y}\mathbf{W}_{QK}\mathbf{Y}^{\top})$, where $\mathbf{A}$ is lower triangular and each row sums to $1$[1]. In addition to the attention weights,

---

[1]$\mathbf{W}_{QK} = \mathbf{w}_Q \mathbf{w}_K^{\top}/\sqrt{d_K}$, where $\mathbf{W}_Q$ and $\mathbf{W}_K$ are typically learned explicitly, and $d_K$ is the number of columns.

*Table 1.* Main notations used in the analogy between attention in GPT and SCM. The first set of symbols represents entities in GPT, and the second set represents entities in SCM.

| Symbol | Description |
|---|---|
| $\boldsymbol{Z}_i$ | output embedding of input symbol $i$, $\boldsymbol{Z}_i \equiv \mathbf{Z}(i, \cdot)$, in attention layer |
| $\boldsymbol{V}_i$ | value vector corresponding to input $i$, $\boldsymbol{V}_i \equiv \mathbf{V}(i, \cdot)$, in attention layer |
| $\mathbf{A}$ | attention matrix |
| $\mathcal{T}$ | Transformer neural network |
| $\mathbf{W}_V, \mathbf{W}_{QK}$ | learnable weight matrices in GPT |
| $X_i$ | a random variable representing node $i$ in an SCM |
| $U_i$ | latent exogenous random variable $i$ in an SCM |
| $\mathbf{G}$ | weighted adjacency matrix of an SCM |
| $\mathcal{G}$ | causal graph structure |

the attention layer calculates a value matrix, $\mathbf{V} = \mathbf{Y}\mathbf{W}_V$, where row $\mathbf{V}(i, \cdot)$ is the value vector of the $i$-th token. Then, the output embeddings are

$$\mathbf{Z} = \mathbf{A}\mathbf{V}, \qquad (1)$$

where the $i$-th row, $\boldsymbol{Z}_i$, is the embedding of the $i$-th output token. In a GPT, several attention layers are stacked and pre-trained such that the $i$-th output embedding in the last layer predicts the $(i+1)$-th input token. That is, it predicts the next token in the sequence.

It is important to note that, in the GPT architecture, the embedding of one token is influenced by another token only by the attention matrix, $\mathbf{A}$. In addition, note that an attention matrix $\mathbf{A}$ is estimated *uniquely* for each input sequence of tokens, using weight matrices $\{\mathbf{W}_{QK}, \mathbf{W}_V\}$ that are learned *commonly* for all in-distribution input sequences.

### 3.2. Structural Causal Model

A structural causal model (SCM) is a model that can encode causal mechanisms in a domain (Pearl, 2009; Spirtes et al., 2000; Peters et al., 2017) and explain data samples generated from these causal mechanisms (Pearl & Mackenzie, 2018). An SCM is a tuple $\{\boldsymbol{U}, \boldsymbol{X}, \mathcal{F}, P(\boldsymbol{U})\}$, where $\boldsymbol{U} = \{U_1, \ldots, U_m\}$ is a set of latent exogenous random variables, $\boldsymbol{X} = \{X_1, \ldots, X_n\}$ is a set of endogenous random variables, $\mathcal{F} = \{f_1, \ldots, f_n\}$ is a set of deterministic functions describing the values $\boldsymbol{X}$ given their direct causes, and $P(\boldsymbol{U})$ is the distribution over $\boldsymbol{U}$. Moreover, each endogenous variable $X_i$ has exactly one unique exogenous cause $U_i$ ($m = n$). The value of an endogenous variable $X_i, \forall i \in [1, \ldots, n]$ is determined by

$$X_i \leftarrow f_i(\boldsymbol{Pa}_i, U_i), \qquad (2)$$

where $\boldsymbol{Pa}_i$ is the set of direct causes (parents in the causal graph) of $X_i$, and left-arrow indicates assignment resulting from the cause-effect relation. A graph $\mathcal{G}$ corresponding

to an SCM consists of one node per variable, and directed edges representing direct causal relations evident from $\mathcal{F}$.

In this paper, we relate the *linear* inter-token relations in GPT attention (Equation 1) to a corresponding linear-Gaussian SCM having directed acyclic graphs (DAG). In these SCM models, each variable is determined by a linear combination of its direct causes and an independently distributed additive noise represented by a corresponding normally distributed exogenous variable. For a linear-Gaussian SCM, let $\mathbf{G}$ be a weight matrix, where $\mathbf{G}(i, j)$ is the weight of the parent (direct cause) node $X_j$ linearly determining the child (direct effect) node $X_i$. Node $X_k$ is not a parent of $X_i$ if and only if $\mathbf{G}(i, k) = 0$. In addition, $\boldsymbol{U} \sim \mathcal{N}(\boldsymbol{\mu_U}, \mathbf{C_U})$, where in this paper, we assume $\mathbf{C_U}$ is a diagonal matrix. The set of functions $\mathcal{F}$ is defined such that $\forall i \in [1, \ldots, n]$,

$$X_i \leftarrow \mathbf{G}(i, \cdot)\boldsymbol{X} + U_i. \qquad (3)$$

Assuming a DAG and causally sorted nodes (ancestors precede their descendants), $\mathbf{G}$ is strictly lower triangular (zeros on the diagonal). Given the assignment, we can write in matrix form $\boldsymbol{X} = \mathbf{G}\boldsymbol{X} + \boldsymbol{U}$, and

$$\boldsymbol{X} = (\mathbf{I} - \mathbf{G})^{-1}\boldsymbol{U}. \qquad (4)$$

As $\mathbf{G}$ is a *strictly* lower-triangular weight matrix, $(\mathbf{I} - \mathbf{G})^{-1}$ is a lower *uni-triangular* matrix (ones on the diagonal). Note that this is equal to the sum of a geometric series

$$(\mathbf{I} - \mathbf{G})^{-1} = \sum_{k=0}^{n-1} \mathbf{G}^k. \qquad (5)$$

It can be seen that element $(i, j)$ represents the cumulative effect of $X_j$ on $X_i$ via all directed paths of length up to $n - 1$. The equivalent weight of a directed path from $X_j$ to $X_i$ is the product of the weights of all edges along that path. The cumulative effect is the sum of the equivalent weights of distinct directed paths from $X_j$ to $X_i$. Note that even if some of the nodes are latent confounders, $(\mathbf{I} - \mathbf{G})^{-1}$ is still

triangular because, by definition, latent confounders have no ancestors and precede other nodes in a causal ordering. Equation 4 represents a system with input $U$, output $X$, and weights $(I - G)^{-1}$. The covariance matrix of the output is

$$
\begin{aligned}
\mathbf{C}_X &= \mathbb{E}[(X - \boldsymbol{\mu}_X)(X - \boldsymbol{\mu}_X)^\top] = \\
&= \mathbb{E}[(\mathbf{I} - \mathbf{G})^{-1}\, \hat{U}\hat{U}^\top\, ((\mathbf{I} - \mathbf{G})^{-1})^\top] = \\
&= [(\mathbf{I} - \mathbf{G})^{-1}]\, \mathbb{E}[\hat{U}\hat{U}^\top]\, [(\mathbf{I} - \mathbf{G})^{-1}]^\top = \\
&= [(\mathbf{I} - \mathbf{G})^{-1}]\, \mathbf{C}_U\, [(\mathbf{I} - \mathbf{G})^{-1}]^\top,
\end{aligned}
\tag{6}
$$

where $\hat{U} = U - \boldsymbol{\mu}_U$ and $\boldsymbol{\mu}_X = (\mathbf{I} - \mathbf{G})^{-1}\boldsymbol{\mu}_U$.

In this paper, we employ a constraint-based causal discovery approach (Spirtes et al., 2000) that uses conditional independence (CI) tests to learn the underlying causal graph. This approach generally requires assuming the causal Markov property and faithfulness.

**Definition 3.1** (Causal Markov). In a causally Markov graph, a variable is independent of all other variables, except its effects, conditional on all its direct causes.

**Definition 3.2** (Faithfulness). A distribution is faithful to a graph if and only if every independence relation true in the distribution is entailed by the graph.

## 4. A Causal Interpretation of GPT

We first describe a relation between GPT and SCM. Then, we present an efficient method for zero-shot causal structure learning—in the presence of latent confounders—for a given input sequence, using a modified version of the ICD algorithm (Rohekar et al., 2021). Finally, we introduce a confidence scoring function for learned causal structures that uses $p$-values computed during causal discovery.

### 4.1. A Relation between GPT and SCM World Model

Rohekar et al. (2024) derived a causal interpretation of BERT (Devlin et al., 2019). We follow a similar approach, with several important modifications and extensions, to derive an SCM-based causal interpretation of GPT. The derived relation between GPT and SCM is threefold (Figure 3):

1. 'Values Matrix' as instances of SCM exogenous nodes,

2. output embeddings as observations of SCM endogenous nodes, and

3. attention matrix as a transitive closure of the SCM graph.

First, unlike BERT-based models, which are pre-trained to predict masked tokens within the input sequence using the surrounding tokens (Devlin et al., 2019), GPT is pre-trained to predict the next tokens in the sequence. That

is, given an input sequence of tokens, $\{t_0, \ldots, t_{n-1}\}$, GPT predicts tokens $\{\hat{t}_1, \ldots, \hat{t}_n\}$. An attention matrix $\mathbf{A}$ and the corresponding values matrix $\mathbf{V}$ have $n$ rows corresponding to input tokens $\{t_0, \ldots, t_{n-1}\}$, and the output embeddings of these tokens are the rows of matrix $\mathbf{Z} = \mathbf{A}\mathbf{V}$. Note that $\mathbf{V} = \mathbf{Y}\mathbf{W}_V$, where $\mathbf{W}_V$ is a weight matrix fixed for all input sequences, and $\mathbf{Y}$ is the input embedding of the tokens in a specific sequence. Each column of $\mathbf{W}_V$ can be viewed as an independent vector onto which the input embeddings are projected. That is, $\mathbf{V}(i, j)$ is the projection of the input embedding of token $t_i$, $\mathbf{Y}(i, \cdot)$, onto the vector $\mathbf{W}_V(\cdot, j)$, which is common to all in-distribution sequences. At inference, each attention matrix of the last attention layer, $\mathbf{A}$, is extracted and a lower uni-triangular matrix is calculated, $\mathbf{D}^{-1}\mathbf{A}$, where $\mathbf{D} \equiv \mathrm{diag}(\mathbf{A})$. Then the covariance matrix is estimated

$$
\mathbf{C} = [\mathbf{D}^{-1}\mathbf{A}][\mathbf{D}^{-1}\mathbf{A}]^\top.
\tag{7}
$$

Note that, unlike Rohekar et al. (2024), who proposed $\mathbf{C} = \mathbf{A}\mathbf{A}^\top$ for *unmasked* self-attention, we utilize the triangular form of masked attention in GPT to revert the attention normalization performed by the softmax and obtain a uni-triangular form. Thus, this covariance matrix allows us to treat properties calculated from different attention matrices in a similar manner. In this paper (Section 4.2 and Section 4.3), the properties we calculate are based on $p$-values from tests of conditional independence between tokens, estimated from the covariance matrix. Next, following Rohekar et al. (2024), we relate each token to an endogenous node in an SCM, and assume $\mathbf{C}_U = \mathbf{I}$ from the central limit theorem. Thus, we equate the covariance $\mathbf{C} = \mathbf{C}_U$:

$$
[\mathbf{D}^{-1}\mathbf{A}][\mathbf{D}^{-1}\mathbf{A}]^\top = [(\mathbf{I} - \mathbf{G})^{-1}][(\mathbf{I} - \mathbf{G})^{-1}]^\top, \tag{8}
$$

where both $\mathbf{D}^{-1}\mathbf{A}$ and $(\mathbf{I} - \mathbf{G})^{-1}$ are lower uni-triangular matrices. The $(i, j)$ elements, $\forall i > j$, of these matrices have the same meaning: influence of token/node $j$ on token/node $i$. Finally, since GPT is pre-trained to predict tokens $\{t_1, \ldots, t_n\}$ given input tokens $\{t_0, \ldots, t_{n-1}\}$, and since the only cross-token influence on embeddings is through the attention matrix, the last attention layer captures the causal structure underlying the output tokens. Earlier attention layers transform embeddings of $\{t_0, \ldots, t_{n-1}\}$ to values, $\mathbf{V}$, which are equivalent to instantiations of the exogenous variables, $U$, in SCM. This follows from equating Equation 1 and Equation 4, where $\mathbf{D}^{-1}\mathbf{A} = (\mathbf{I} - \mathbf{G})^{-1}$. That is, we equate the outputs: tokens' embeddings and SCM nodes' values. If some of the nodes are hidden confounders, then the corresponding rows and columns are removed,

$$
\mathbf{D}^{-1}\mathbf{A} = [(\mathbf{I} - \mathbf{G})^{-1}]_{\hat{i},\hat{i}},
\tag{9}
$$

where $i$ denotes the indices of nodes hidden in the world model ($\hat{i}$ denotes the omission of the corresponding rows and columns).

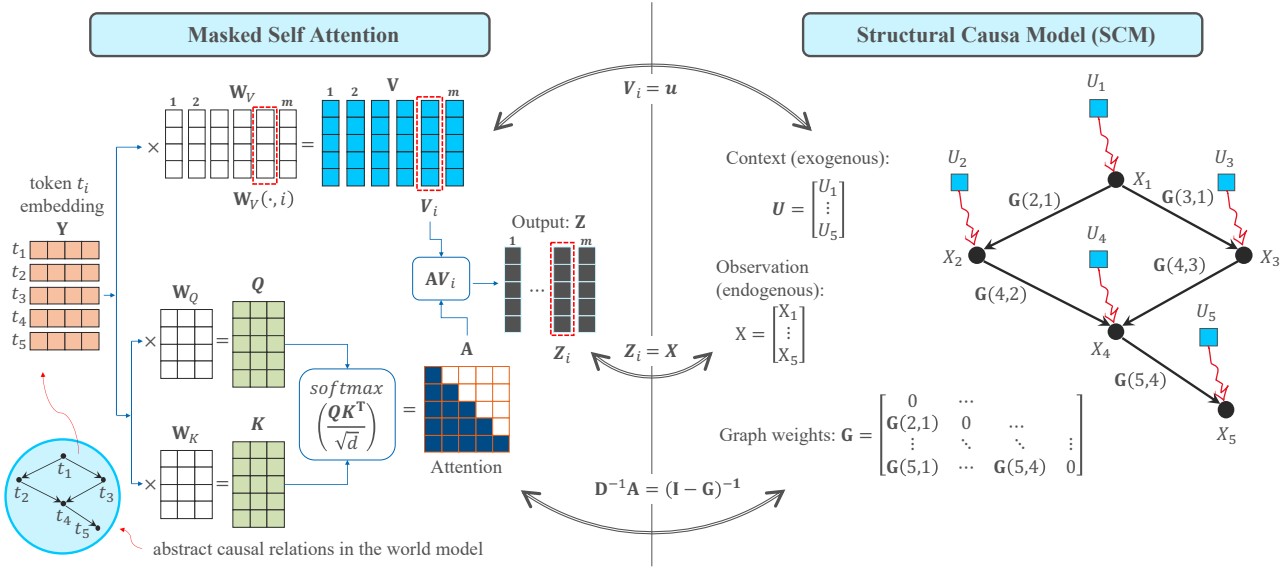

*Figure 3.* Relations between GPT (left) and SCM (right), derived in Section 4.1. **'Values matrix' as exogenous variables in SCM**: In the attention mechanism, the input embeddings matrix $\mathbf{Y}$ is multiplied by the column vectors $\mathbf{W}_V(\cdot, i)$ of the weight matrix $\mathbf{W}_V$ to form the column vectors $\mathbf{V}_i$ of the values matrix $\mathbf{V}$. Each values vector $\mathbf{V}_i$ is treated as an instantiation $u$ of the exogenous nodes in SCM, where element $j$ in $\mathbf{V}_i$ is an instantiation of node $U_j$. **Output embeddings as observed nodes**: Each values vector $\mathbf{V}_i$ is multiplied by the attention matrix, resulting in a column vector $\mathbf{Z}_i$ of the output embedding. This corresponds to an observation of the endogenous nodes in SCM. **Attention matrix as a transitive closure of a causal graph**: An element $\mathbf{A}(i, j)$ in the attention matrix reflects the 'attention' given to token $j$ when computing the embedding of token $i$. This corresponds to the influence that node $j$ has on node $i$ through all directed paths in the causal graph, as estimated by $(\mathbf{I} - \mathbf{G})^{-1} = \sum_k \mathbf{G}^k$. If some nodes in SCM are hidden confounders, then the attention matrix reflects $[(\mathbf{I} - \mathbf{G})^{-1}]$ after removing the rows and columns corresponding to the hidden nodes.

In light of the causal interpretation of GPT, one important question is what *causal world model* the GPT architecture supports. Note that GPT's non-linear transformations do not affect inter-token relations. Often, a single causal structure is assumed to govern a domain. In contrast, the causal world model entailed by the causal interpretation of GPT assumes a distinct SCM for each sequence. Specifically, in a causal world model supported by a GPT with $k$ heads in the last attention layer, each sequence is assumed to be generated by an ensemble of $k$ SCMs.

In addition, for a given head, the causal structure over a sequence of tokens $\{t_1, \ldots, t_n\}$ is identical to the corresponding subgraph over these tokens in all in-distribution extensions of the sequence. That is, given a sequence of tokens $\{t_1, \ldots, t_n\}$ and a corresponding graph structure $\mathcal{G}_n$, observing any next token $t_{n+1}$, such that $\{t_1, \ldots, t_n, t_{n+1}\}$ is in-distribution, should not violate the causal relations in $\mathcal{G}_n$ and may only reveal relations between tokens $\{t_1, \ldots, t_n\}$ and token $t_{n+1}$.

### 4.2. GPT for Zero-Shot Causal Structure Learning

The causal interpretation presented in this paper leads to a view in which each attention module captures associations (correlations) between input tokens that are induced

by the underlying causal structure. Although this supports only rung-1 inference in the ladder of causation (Pearl & Mackenzie, 2018) many of the underlying causal relations can be extracted under certain assumptions—even in the presence of latent confounders and selection bias (Spirtes et al., 2000). These relations are generally represented in a type of causal structure known as a partial ancestral graph (PAG) (Richardson & Spirtes, 2002). We follow a procedure called ABCD, proposed by Rohekar et al. (2024), with several modifications. First, since the causal (topological) order is given (restricted by the masked attention in GPT), we can apply causal discovery recursively to efficiently learn the causal structure. To this end, we slightly modify the iterative causal discovery (ICD) algorithm (Rohekar et al., 2021), as described in Appendix B, to reconstruct a causal structure at each recursive iteration. The procedure is outlined in Algorithm 2. The input is a sequence of tokens over which we construct the graph. The output is a PAG structure. In line 2, an exit condition corresponding to the base case (a single-node graph) is tested. In line 3, the last token is popped from the sequence and assigned to $t_n$, resulting in a shorter sequence $\mathbf{S}'$. Then, a recursive call is made in line 4 to learn the structure over the tokens in $\mathbf{S}'$. Note that since it is ensured that $t_n$ is not an ancestor of any token in $\mathbf{S}'$, the skeleton and v-structure relations of $\mathcal{G}'$ are guaranteed

not to change when $t_n$ is added back to the graph (Spirtes et al., 2000). In lines 5–7, token $t_n$ is connected to every node in $\mathcal{G}'$. Finally, in line 8, edges between $t_n$ and the rest of the graph are learned (removed if conditional independence is found) using the ICD algorithm (Rohekar et al., 2021) and the graph is oriented (Zhang, 2008). Although we use ICD, other constraint-based causal discovery algorithms (Colombo et al., 2012; Claassen et al., 2013; Yehezkel & Lerner, 2009; Spirtes et al., 2000; Rohekar et al., 2018; Nisimov et al., 2021), differing in their underlying assumptions, can also be used.

---

**Algorithm 1:** Recursive Causal Discovery for GPT

---

**Input:** $\boldsymbol{S}$: a sequence of tokens $\{t_1, \ldots, t_n\}$

**Output:** $\mathcal{G}$: a partial ancestral graph (PAG)

1   **Function** `LearnStructure(`$\boldsymbol{S}$`):`
2     if $|\boldsymbol{S}| = 1$ then return a graph with the single node in $\boldsymbol{S}$
3     $t_n, \boldsymbol{S}' \leftarrow \text{pop}(\boldsymbol{S})$
4     $\mathcal{G}' \leftarrow \text{LearnStructure}(\boldsymbol{S}')$
5     $\mathcal{G} \leftarrow \mathcal{G}' + \{t_n\}$
6     set $\boldsymbol{E}$ to the set of edges (circle edge-marks) between $t_n$ and every node in $\mathcal{G}'$
7     connect $\boldsymbol{E}$ in $\mathcal{G}$
8     test CI for edges in $\boldsymbol{E}$ and orient $\mathcal{G}$ using Algorithm 3 (Appendix B)
9     return $\mathcal{G}$

---

Thus, a causal structure for a particular output sequence can be inferred in a zero-shot manner directly from the attention matrix in the last layer. In multi-head attention, the final attention layer, having $k$ heads, is the last layer in which tokens may affect one another. Hence, Algorithm 2 is invoked independently for each head, returning a set of $k$ structures.

### 4.3. Causal Structure Confidence

In this section, we derive a metric that describes how compatible a sequence is with the causal model implicitly encoded by GPT. Given an output sequence of tokens, $\boldsymbol{S}$, and a causal structure $\mathcal{G}$ recovered from the last attention layer $\mathbf{A}$, can we score the confidence in this causal structure? Recall that in the proposed world model, each sequence has its own causal structure, and each causal structure may include latent variables. Since it is unclear how to calculate likelihood $P(\boldsymbol{S} \mid \mathcal{G})$, we propose the following approach.

A causal structure-learning algorithm performs multiple statistical tests of conditional independence (CI) using the covariance matrix estimated from the attention matrix. These CI tests calculate $p$-values and compare them against a pre-

determined significance threshold ($\alpha$). It is important to note that a causal structure can be uniquely represented by a set of CI tests and their results. Hence, we propose a scoring function based on the distribution of these $p$-values to evaluate the confidence in a structure learned from a given attention matrix. A complete undirected graph corresponds to a lack of knowledge about causal relations. Generally, causal structure-learning algorithms prune edges from this graph based on statistical CI tests between pairs of variables (tokens, in our case). The removal of edges between independent variables may then entail causal relations between other variables (Zhang, 2008).

Let $\boldsymbol{p} = \{p_1, \ldots, p_\ell\}$ be the set of all $p$-values computed as part of causal structure learning. The null hypothesis corresponds to independence, where $p$-values greater than the significance threshold, $\alpha$, correspond to edges removed from the complete graph. We define $\boldsymbol{p}_{\text{ind}} = \{p \in \boldsymbol{p} \mid p \geq \alpha\}$, and $\boldsymbol{p}_{\text{dep}} = \{p \in \boldsymbol{p} \mid p < \alpha\}$. Since $p$-values are uniformly distributed under the null hypothesis, we expect the entropy of $p$-values corresponding to independence, $H_{\text{ind}}$, to be higher for matrices that correspond to a structure than for those that do not. Conversely, we expect the distribution of $\boldsymbol{p}_{\text{dep}}$ to be weighted toward zero. Hence, the entropy of $p$-values corresponding to dependence relations, $H_{\text{dep}}$, is expected to be lower for matrices that correspond to a structure compared to those that do not. We therefore define the following confidence score, given an attention matrix $\mathbf{A}$:

$$R(\mathbf{A}) = H_{\text{ind}} - H_{\text{dep}}, \tag{10}$$

which captures the contrast between dependence and independence relations entailed by the learned causal graph.

## 5. Experiments and Results

We use an experimental framework in which the world layout and rules governing the generation of sequences are well defined and known, but are not utilized during training. We measure how well attention in the trained GPT model represents a causal world model and whether this representation is correlated with the ability to generate tokens that adhere to the world rules.

### 5.1. Setup

We used two controlled environments: Othello and Chess strategy games. For Othello, we examined a GPT model trained by Li et al. (2023) on $\sim$132 thousand real-world sequences, and for Chess, we examined a GPT model trained by Toshniwal et al. (2022) on $\sim$2.9 million real-world games.

For both environments, no information about the game board layout or game rules was used during their training process, and the training data consisted of games in which

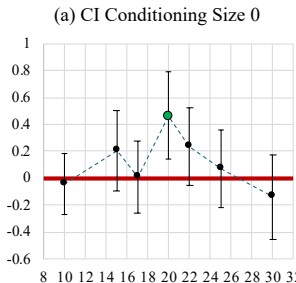 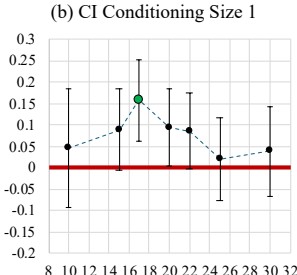 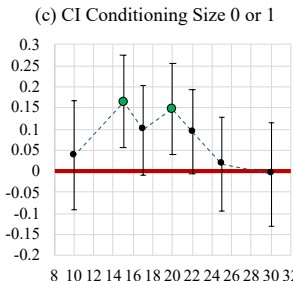 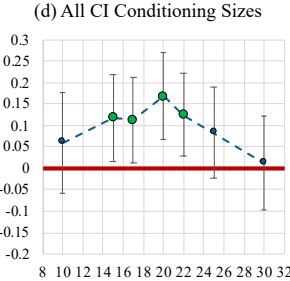

*Figure 4.* Average difference in structural confidence between legal and illegal move generation (vertical axis) for different input-sequence lengths (horizontal axis). Error bars represent the 95% confidence interval calculated using a t-test. Confidence scores are calculated from $p$-values of: (a) all unconditional (marginal) independence tests, (b) all CI tests having exactly one conditioning node, (c) only tests from both cases (a) and (b), and (d) only CI tests without limiting the conditioning set sizes, needed to reconstruct a causal structure.

players played with the intention of winning. For example, positional encoding was not used.

In all our experiments, we used test sets that are out-of-distribution with respect to the training set, consisting of sequences of randomly sampled legal moves (not by the GPT models), lacking the objective of winning. In other words, the support of the test distribution is not a subset of the support of the training distribution, where $\text{supp}(P_{\text{train}}) \subset \text{supp}(P_{\text{test}})$. See Appendix A for an empirical comparison between the sets. This enables evaluating whether the model implicitly encodes the game rules. For both Chess and Othello, test sets consisted of 1,000 randomly generated sequences of legal moves. See Appendix A for more details.

For causal discovery implementation and empirical evaluation we used the Causality Lab repository: `github.com/IntelLabs/causality-lab`.

### 5.2. Ablation Study

We examine legal move generation with respect to 1) limiting the condition set sizes in the CI tests used to learn causal structures, and 2) pruning attention heads based on the confidence scores of their corresponding causal structures.

#### 5.2.1. CONTRIBUTION OF CI TESTS

We examine whether conditional independence (CI) tests from which the causal structure is entailed provide an advantage over pairwise correlations directly represented by elements in the attention matrix. To this end, we calculate the confidence score (Equation 10) using $p$-values from: a) all pairwise marginal independence relations (from raw attention-matrix elements)—CI conditioning size 0; b) CI tests having exactly one node in the conditioning set; c) all CI tests having either an empty or single-node conditioning set; and d) all CI tests used to reconstruct the causal

structure without limiting conditioning set sizes. The results are shown in Figure 4. Let $\bar{R}_{\text{legal}}$ be the average structural confidence score of sequences for which a legal token was generated, and $\bar{R}_{\text{illegal}}$ be the average structural confidence score of sequences for which an illegal token was generated. The vertical axis represents the difference in structural confidence scores $\bar{R}_{\text{legal}} - \bar{R}_{\text{illegal}}$. Error bars indicate 95% confidence intervals (unpaired t-test). The horizontal axis indicates sequence length.

It is evident that when relying solely on raw attention values, case (a), the difference between legal and illegal generated tokens is not statistically significant, except for sequence length 20. Relying solely on CI-test with exactly one node in the conditioning set, case (b), the difference in structural confidence is positive for all tested sequence lengths, but statistically significant only for sequence length 17. When employing pairwise correlations and CI tests with exactly one node in the conditioning tests, case (c), the result is statistically significant for both sequence lengths 17 and 20, implying that these two types of tests are complementary. Finally, using all CI-tests needed to learn the causal graph, without limiting the conditioning set sizes, case (d), provides the best results: sequence lengths in range $[15, 22]$ are statistically significant, and the difference between legal and illegal scores is positive ($\bar{R}_{\text{legal}} > \bar{R}_{\text{illegal}}$) for all tested sequence lengths.

#### 5.2.2. ATTENTION HEADS PRUNING BASED ON CONFIDENCE SCORE

In this experiment, we examine the importance of each attention head (in multi-head attention) for legal-move generation. We evaluate the importance of a head by the degree of confidence with which it represents a causal structure. This is different from the experiments in Section 5.3 and Section 5.2.1 where the average structural confidence score of the heads was associated with each test sequence.

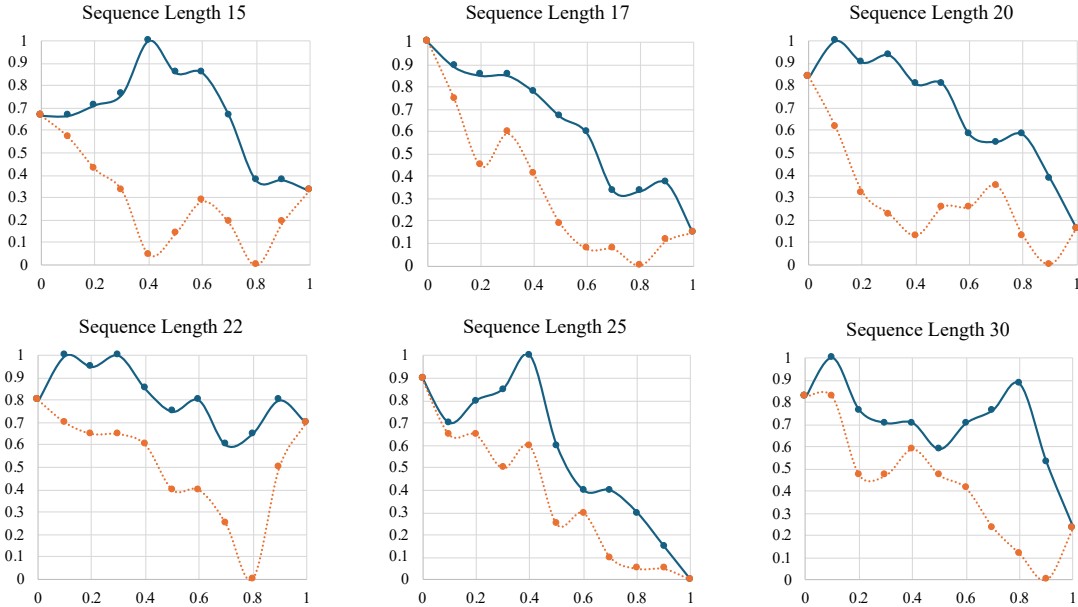

*Figure 5.* Normalized accuracy of legal-move generation (vertical axis) as a function of the percentage of heads pruned (horizontal axis) based on structural confidence. A solid blue curve represents pruning a percentage of heads having the lowest structural confidence, while a dotted orange curve represents pruning in the reverse order (pruning a percentage of heads having the highest structural confidence).

Here, a structural confidence score is calculated for each attention head for each sequence in the test set. That is, for 1,000 test sequences and 8 heads in the last attention layer, there is a set of 8,000 scores. This set, denoted $R$, is sorted in ascending order. From this sorted set, nine equally spaced values are selected as thresholds, denoted $th = \{th_1, \ldots, th_9\}$, corresponding to the 10%, 20%, ..., 90% percentiles. Given a threshold $th_i$, for each test sequence the attention heads that have structural confidence scores lower than the threshold are pruned (skipped in the forward pass) and the next token is generated without those heads. Hence, the number of pruned heads may vary from sequence to sequence. We then calculate the legal-move generation accuracy for each threshold, that is, accuracy per pruning percentile. Note that retraining the model after pruning is not required (Voita et al., 2019).

In our case, it is expected that pruning heads with low structural confidence will have limited impact on the accuracy. To examine this, we compare the accuracy to that of a *reverse-order pruning* process. In this process, we prune heads having high structural confidence scores while keeping those with lower scores. Specifically, we sort the set of scores, $R$, in a descending order, and for each threshold, prune the heads that have higher structural confidence scores. Under the assumption that GPT implicitly uses a causal world model to generate the next tokens, we expect that pruning heads having low structural confidence scores will result in higher legal-move accuracy and larger area under curve

(accuracy as a function of pruning percentile) than in the reverse-order pruning process.

In Figure 5, it is evident that pruning heads with lower structural confidence scores (solid blue curve) results in higher legal-move generation accuracy and greater area under curve, compared to removing heads with higher structural confidence scores (dotted orange curve). This demonstrates the importance of individual attention heads that encode structural information for generating legal moves.

### 5.3. Legal Move Generation vs. Structural Confidence

Is there a relation between generating legal tokens (moves) and how well attention matrices implicitly represent causal structures? Recall that the model was not trained explicitly to generate legal game moves but rather to predict the next move played by a human with the intention of winning the game. Moreover, information about the game, such as the existence of a board game and rules, were not provided to the model (Li et al., 2023; Toshniwal et al., 2022).

In this experiment, we examine whether the cases in which the model generates illegal tokens are also cases where the causal structure is less distinctive, as measured by the structural confidence score, $R$ (Equation 10). Here, the score for a given sequence is the average of structural confidence scores calculated for the attention heads in the last layer. Recall that structural confidence is not an objective in GPT pretraining.

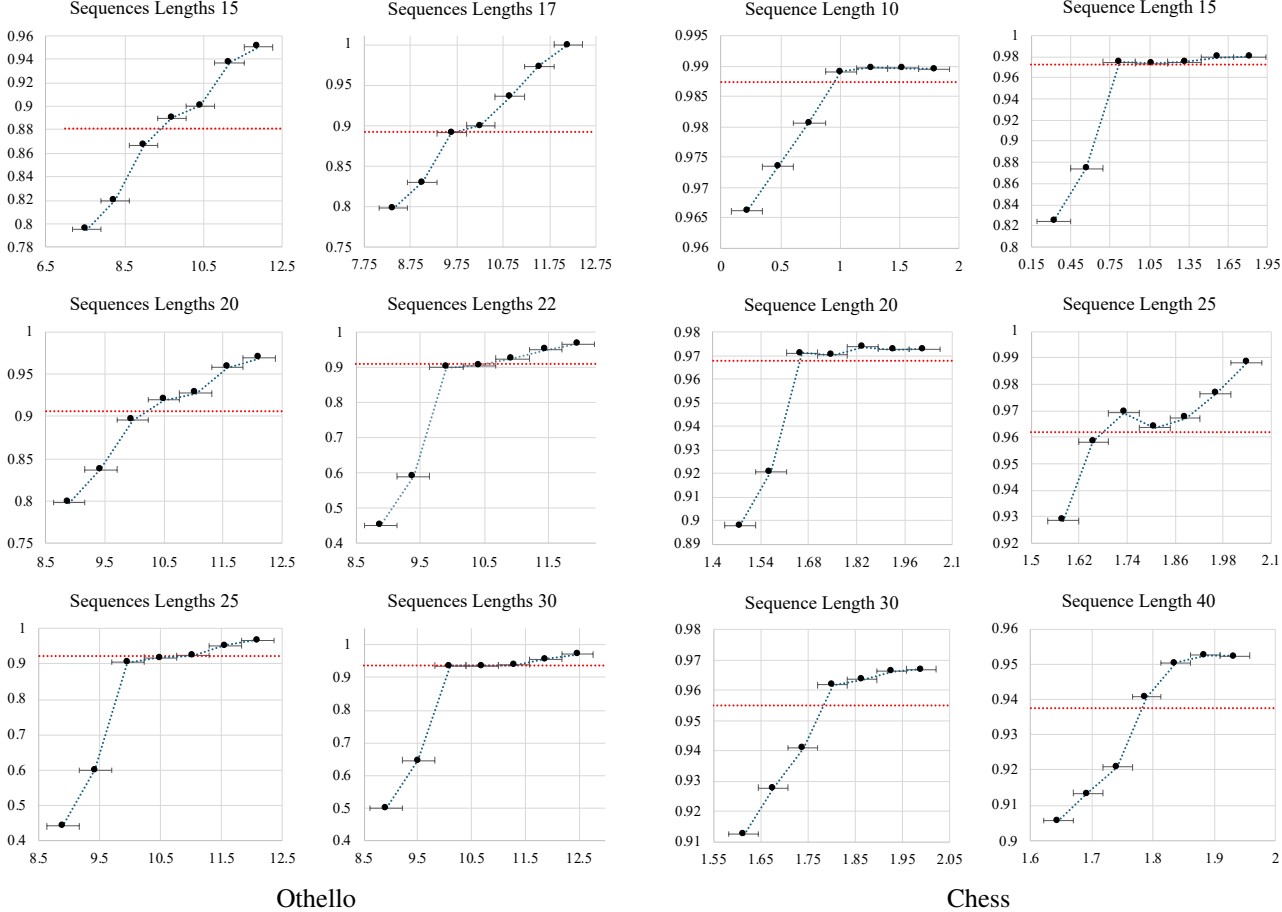

Othello — Chess

*Figure 6.* Legal move generation accuracy (vertical axis) as a function of structural confidence score $R$ (horizontal axis) for Othello (left two columns) and Chess (right two columns). Horizontal limits for each point indicate interval of $R$ in which accuracy was averaged. Horizontal dotted red line represents average accuracy. Accuracy increases with the structural confidence score.

From Figure 6, for Othello and Chess, it is evident that the legal move generation accuracy (vertical axis) increases with the structural confidence score $R$ (horizontal axis). That is, GPT is more likely to generate legal tokens for out-of-distribution inputs when a causal structure can be learned more confidently from its attention maps.

## 6. Conclusions

We presented a causal interpretation of GPT that may clarify the apparent emergence of world models in recent studies and extend their findings. Following this interpretation, we described a method that utilizes the triangular form of the attention matrices in GPT to recover the covariance matrix of SCM endogenous nodes, and efficiently learn the causal graphs for input sequences in a zero-shot manner. Furthermore, we introduced a confidence scoring function for the learned graphs, based on the difference in entropy between the dependence and independence populations of $p$-values. Finally, using the controlled environments of the

Othello and Chess strategy games, we demonstrated that GPT implicitly learns to represent causal structures in attention heads. Specifically, in cases where the confidence in recovering structures from the attention matrices is low, GPT generally fails to generate a token that adheres to the game rules. In future work, these results may provide insights into the sources of hallucination in GPT-based models and methods for detecting them.

## Impact statement

We propose a link between the internal mechanism of the GPT model and its ability to implicitly encode the world model of a given domain. As GPT models become increasingly widespread, it is crucial to understand the reasoning behind their outputs in relation to domain-specific rules. This understanding enables better human supervision and oversight of these complex automated models. We believe our work has positive societal implications by fostering transparency and accountability in AI-driven decision-making.

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

## A. Comparison between Training and Test Data

The data used to train the GPT model consisted of real-world sequences of game moves (Li et al., 2023). These moves were played strategically with the intention of winning the game. In contrast, the experiments in the paper were conducted using test data consisting of randomly generated sequences of moves that adhered to the game rules, without considering the outcome of the game.

### A.1. Accuracy in Predicting the Legal Next Move for Test Sequences

In Figure 7, we plot the accuracy of the model in generating a legal next move (vertical axis) in Othello and Chess as a function of the number of moves (sequence length) in the test input sequences (horizontal axis). Note that the test sequences were not generated by the GPT model. Instead, each move in a test sequence is sampled uniformly from the set of next legal moves, according to the game rules.

For Othello, note that length-$n$ sequences are test sequences that are trimmed to keep only the first $n$ tokens, such that the same 1,000 sequences are used for all evaluated lengths. Although the average accuracy of the model is 95% (dashed red line), it is not uniformly distributed across different sequence lengths. For example, given a sequence of 15 moves, GPT generates a legal 16th move 88% of the time (adhering to the game board state and rules). It is evident that the accuracy is significantly lower for input sequence lengths in the range $[10, 30]$ (below the average of 95%). From the Othello game rules, at the beginning of a game there are only four legal moves, and as the game unfolds, the number of possible legal moves generally increases before finally decreasing again as the number of vacant spaces on the board diminishes. It might be that memorization of surface-level statistics can take place at the beginning of the game. We therefore report experimental results for input sequences with sizes in the range $[10, 30]$ (gray area), where the accuracy is lower than average. Throughout the experiments, we employ Algorithm 2 for causal discovery using partial correlation with a significance level of $\alpha = 0.01$ for testing conditional independence (CI tests).

For Chess, to avoid the possibility of memorization, we use sequences having at least 10 moves. Then, since the accuracy of the model constantly decreases with the sequence length, we use sequences up to 40 moves. Longer sequences generally lead to game termination before the full sequence length is reached. Due to the small error rate for shorter sequences, in our experiment we used a test set of 10,000 samples for evaluating sequence lengths up to 15 moves.

### A.2. Difference between Train and Test Datasets

Recall that the test sequences were synthesized by sampling each move uniformly from the set of legal next moves according to the game rules. We measure the difference between the distributions of sequences in the training dataset, $\boldsymbol{D}^{\text{train}}$, and the test dataset, $\boldsymbol{D}^{\text{test}}$, by estimating $n$-gram frequencies. For a given sequence, $\{t_0, \ldots, t_{\ell-1}\}$, we extract the last $n$ tokens, assuming that the probability of the next generated token $t_\ell$ depends only on these $n$ tokens,

$$P(t_\ell | t_0, \ldots, t_{\ell-1}) = P(t_\ell | t_{\ell-n}, \ldots, t_{\ell-1}). \tag{11}$$

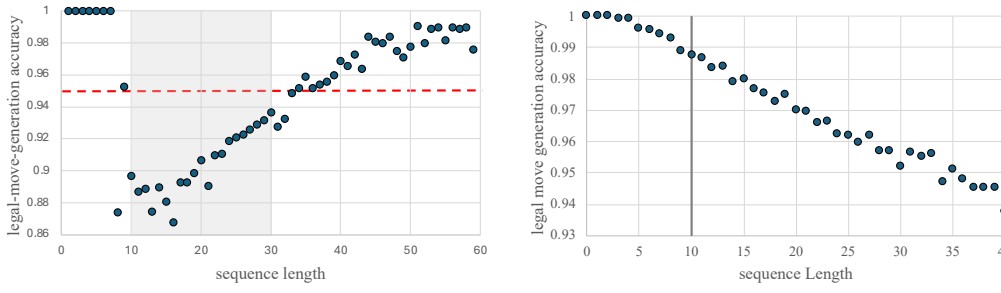

*Figure 7.* Baseline model accuracy of generating legal Othello (left) and Chess (right) game moves. Models were trained by Li et al. (2023) for Othello and by Toshniwal et al. (2022) for Chess on real-world games to predict the next move. The test set consists of randomly generated sequences of legal moves. Measured accuracy: the percentage of generated moves that are legal according to the game rules. The gray area for Othello highlights input sequences with sizes in the range $[10, 30]$, where the accuracy is lower than the average of 95% (red dashed line). For Chess, the input sequences that are considered have 10 or more moves (gray line threshold).

For the $i$-th sequence in the test set, trimmed to length $\ell$, we count the number of occurrences, $N_n^{\text{test}|\text{train}}(i)$, of the $n$-gram $\{t_{\ell-n}, \ldots, t_{\ell-1}\}$ of the test sequence in the training data sequences, trimmed to length $\ell$. We then divide this count by the number of training sequences, $|\boldsymbol{D}^{\text{train}}|$, and estimate the mean $\mu_n^{\text{test}}$ ($|\boldsymbol{D}^{\text{test}}|$ is the number of test sequences),

$$\mu_n^{\text{test}|\text{train}} = \frac{1}{|\boldsymbol{D}^{\text{test}}|} \sum_i \frac{N_n^{\text{test}}(i)}{|\boldsymbol{D}^{\text{train}}|}. \tag{12}$$

Similarly, using sequences excluded from the training data, we estimate $\mu_n^{\text{train}|\text{train}}$, the percentage of occurrences of $n$-grams of training sequences in the training data sequences. For each sequence length evaluated in the paper, $\ell \in \{15, 17, 20, 22, 25, 30\}$, we calculate the percentage of $n$-gram occurrences for $n \in [2, \ldots, 6]$. We then compare the percentage of occurrences $\mu_n^{\text{test}|\text{train}}$ and $\mu_n^{\text{train}|\text{train}}$ in Figure 8. This evaluation clearly shows that the distribution of real-world sequences played with the intention of winning ($\boldsymbol{D}^{\text{train}}$) is different from that of randomly generated sequences ($\boldsymbol{D}^{\text{test}}$) used in the paper to examine the trained GPT model.

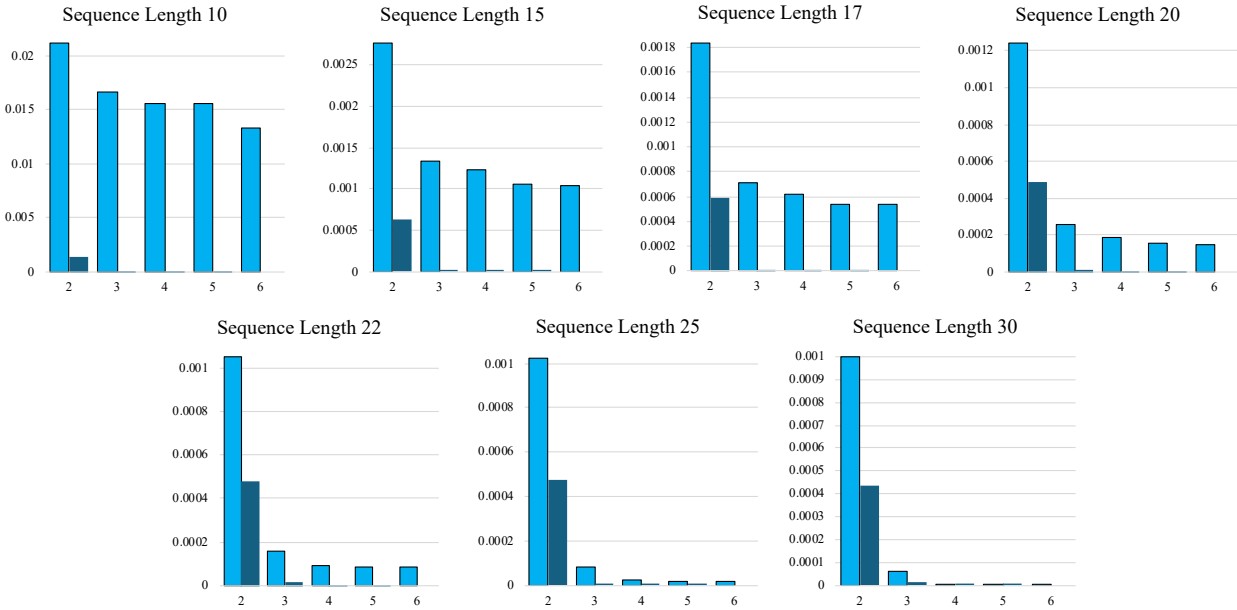

*Figure 8.* Percentage of occurrences (vertical axis) of $n$-grams from test and training sequences in the training data for $n \in [2, \ldots, 6]$ (horizontal axis). Light blue columns are $\mu_n^{\text{train}|\text{train}}$, and dark blue are $\mu_n^{\text{test}|\text{train}}$ values. The clear difference between $\mu_n^{\text{test}|\text{train}}$ and $\mu_n^{\text{train}|\text{train}}$ which indicates a clear difference between the distributions of real-world sequences used to train the GPT model and randomly generated sequences used for evaluation.

## B. Recursive Causal Discovery from GPT Attention

We describe our method in Algorithm 2, where, given an input sequence, a causal structure is learned from an attention matrix in the last layer. In this section, we provide a more detailed explanation of line 8, where the ICD algorithm (Rohekar et al., 2021), modified to learn only a given set of edges, is called. The operations in line 8 are largely similar to those in the ABCD algorithm (Rohekar et al., 2024). The main difference is that this step refines a partially learned causal structure by testing conditional independence between pairs of nodes connected by edges in a given list $E$.

The operations in line 8 of Algorithm 2 are as follows. First, covariance is estimated from an attention matrix $\mathbf{A}$,

$$\mathbf{C} = \left[\mathbf{D}^{-1}\mathbf{A}\right]\left[\mathbf{D}^{-1}\mathbf{A}\right]^{\top}, \tag{13}$$

where $\mathbf{D} \equiv \text{diag}(\mathbf{A})$ is a diagonal matrix consisting of elements on the diagonal of $\mathbf{A}$, such that $\mathbf{D}^{-1}\mathbf{A}$ is a uni-triangular matrix. Then, a correlation matrix is estimated,

$$\mathbf{R} = \text{diag}(\mathbf{C})^{-1/2}\,\mathbf{C}\,\text{diag}(\mathbf{C})^{-1/2}. \tag{14}$$

Conditional independence between two variables $X$ and $Y$, conditioned on set $\boldsymbol{Z}$, is estimated by calculating the partial correlation from $\mathbf{R}$. Then, let $\mathrm{Ind}\,(X, Y | \boldsymbol{Z})$ denote a CI test based on partial correlation, where $p$-values are estimated using Fisher z-transform. Finally, ICD is called to learn a set of edges using $\mathrm{Ind}$.

In Algorithm 3, we provide a simple modification of ICD such that it learns only the edges in $\boldsymbol{E}$ and uses a given initial graph. In red we strike out parts of the ICD and in blue are our additions. The rest of the pseudo-code is exactly as given by Rohekar et al. (2021). As input, we add the initial graph $\mathcal{G}$ to be used and further refined, and add the set of edges $\boldsymbol{E}$ to be learned (remove edges connecting conditionally independent nodes). In line 1, we remove the initialization of a complete graph, since the initial graph is given as input. In line 3 and line 6, we add the set of edges $\boldsymbol{E}$ to be tested within the ICD iteration function. Lastly, in line 8, only edges in $\boldsymbol{E}$, rather than all edges in $\mathcal{G}$, are tested.

Overall, utilizing the causal order enforced by the triangular form of the GPT attention matrix, each recursive call assumes that the current graph is the final learned graph, except for the edges connecting the newly added node to the rest of the graph nodes (edge list $\boldsymbol{E}$). Note that this does not violate the ICD-Sep conditions (Rohekar et al., 2021), which constitute a sufficient set for ensuring a sound and complete causal discovery algorithm. By considering only the edges connecting a node to its predecessors in the given causal order, a significantly lower number of CI tests are required for learning the causal graph compared to the unmodified ICD algorithm.

---

**Algorithm 2:** Causal Discovery for GPT

---

**Input:** $\boldsymbol{S}$: a sequence of tokens $\{t_1, \ldots, t_n\}$

**Output:** $\mathcal{G}$: a partial ancestral graph (PAG)

---

1 **Function** `LearnStructure(`$\boldsymbol{S}$`)`:

2      if $|\boldsymbol{S}| = 1$ then return a graph with the single node in $\boldsymbol{S}$

3      $t_n, \boldsymbol{S}' \leftarrow \mathrm{pop}(\boldsymbol{S})$

4      $\mathcal{G}' \leftarrow$ `LearnStructure(`$\boldsymbol{S}'$`)`

5      $\mathcal{G} \leftarrow \mathcal{G}' + \{t_n\}$

6      set $\boldsymbol{E}$ to the set of edges (circle edge-marks) between $t_n$ and every node in $\mathcal{G}'$

7      connect $\boldsymbol{E}$ in $\mathcal{G}$

8      test CI for edges in $\boldsymbol{E}$ and orient $\mathcal{G}$ using `ICD` (Rohekar et al., 2021)

9      return $\mathcal{G}$

---

**Algorithm 3:** Modified ICD (Rohekar et al., 2021) algorithm

---

**Input:**
    Ind: a conditional independence oracle
    $\mathcal{G}$: initial PAG
    $\mathbf{E}$: set of edges to be learned

**Output:**
    $\mathcal{G}$: a PAG

**1**   initialize: $r \leftarrow 0$, ~~$\mathcal{G} \leftarrow$ a complete graph with 'o' edge-marks,~~ and $done \leftarrow$ False

**2**   **while** $(r \leq n)$ & $(done = \text{False})$ **do**
**3**     $(\mathcal{G}, done) \leftarrow$ Iteration($\mathbf{E}, \mathcal{G}, r$)            ▷ refine $\mathcal{G}$ using conditioning sets of size r
**4**     $r \leftarrow r + 1$
**5**   **return** $\mathcal{G}$

**6**   **Function** Iteration($\mathbf{E}, \mathcal{G}, r$)**:**

**7**     $done \leftarrow$ True
**8**     **for** *edge* $(X, Y)$ *in* $\mathbf{E}$ ~~edges($\mathcal{G}$)~~ **do**
**9**       $\{\mathbf{Z}_i\}_{i=1}^{\ell} \leftarrow$ PDSepRange $(X, Y, \text{r}, \mathcal{G})$          ▷ $\mathbf{Z}_i$ complies with ICD-Sep conditions
**10**       **if** $\ell > 0$ **then**
**11**         $done \leftarrow$ False
**12**         **for** $i \leftarrow 1$ **to** $\ell$ **do**
**13**           **if** $\text{Ind}(X, Y | \mathbf{Z}_i)$ **then**
**14**             remove edge $(X, Y)$ from $\mathcal{G}$
**15**             record $\mathbf{Z}_i$ as a separating set for $(X, Y)$
**16**             **break**

**17**     orient edges in $\mathcal{G}$
**18**     **return** $(\mathcal{G}, done)$

---

