# OpenReview forum: "A Causal World Model Underlying Next Token Prediction: Exploring GPT in a Controlled Environment"
_ICML.cc/2025/Conference — ICML 2025 poster_

### Official Review · Reviewer_g9Sr · 2025-02-24

**Overall Recommendation:** 3

**Summary:**

This paper explores whether GPT models, designed for next-token prediction, implicitly learn a causal world model from which sequences are generated. The authors derive a causal interpretation of the GPT attention mechanism and suggest that GPT models can be used for zero-shot causal structure learning with a proposed confidence score. The study is conducted in a controlled environment using Othello and Chess games, where a GPT model trained on real-world games is tested on synthetic data sequences of random legal moves. Results indicate that the GPT model can often generate legal next moves for out-of-distribution sequences by capturing causal structures encoded in the attention mechanism, but struggles and fails to capture causal structures when generating illegal moves. The introduction elaborates on GPT's unexpected capabilities beyond next-token predictions, proposing that these abilities might arise from implicit learning of causal structures during pre-training due to Occam's razor, which favors simpler, more compact solutions. The paper builds on recent methods for causal interpretation in transformer models, adapting them to GPT, and investigates the correlation between GPT-generated errors and the uncertainty in representing causal structures.

**Claims And Evidence:**

Partly. Please see questions.

**Essential References Not Discussed:**

No to the best of my knowledge.

**Experimental Designs Or Analyses:**

I have checked the soundness of experiments.

**Methods And Evaluation Criteria:**

The proposed method is simple yet effective. However, I have a question about whether the results can only be obtained through causality or if they can also be achieved using correlation-based methods. Please see questions.

**Other Comments Or Suggestions:**

In Eq (9), the LHS is a matrix while the RHS is a value.

The computation of $H_{ind}$ and $H_{dep}$ should be clearly defined in the main text.

**Other Strengths And Weaknesses:**

**Strengths:**

1. This paper is the first to introduce causality into the controlled study of the world model.

2. It is interesting to see that attention weights pruning can improve the legal move, and that legal moves are correlated with confidence.

**Weaknesses:**

1. For the results discussed in Sections 5.2 and 5.3, I would like to know whether a causality-based method is necessary. I will improve the score if the authors can address this point. (Please refer to Questions 1 and 2)

2. The authors state that "they do not explain how the board game is encoded within the attention matrices and why the attention mechanism can represent the board state". But the authors do not explain these questions clearly. (Please refer to Question 3)

**Questions For Authors:**

It would be interesting to explore whether correlation-based methods can reproduce the results presented in Sections 5.2 and 5.3. Some simple correlation-based methods include:

1. **Attention as the "causal structure" $\mathcal{G}$**:
   1. Let $G$ be the adjacency matrix of $\mathcal{G}$. Set $G_{ij} = 1$ if $A_{ij} > \frac{\alpha}{i}$.
   2. Repeat the algorithm in Section 4.3 to compute the confidence.
   3. Check whether the results in Sections 5.2 and 5.3 can be reproduced.

2. **Precision matrix as the "causal structure" $\mathcal{G}$**:
   1. Compute the covariance matrix based on attention $C = [D^{-1}A][D^{-1}A]^T$.
   2. Recover the precision matrix $\Theta$ using an existing library (e.g., [Ref. A]), and set $G_{ij} = 1$ if $\Theta_{ij} > \alpha$. If $C$ is well-conditioned, one can also use $\Theta = C^{-1}$.
   3. Repeat the algorithm in Section 4.3 to compute the confidence.
   4. Check whether the results in Sections 5.2 and 5.3 can be reproduced.

3. The authors use the last layer attention. Why is only the last layer useful? The authors should explain the functions of the previous layers' attentions and MLPs.

[Ref. A] https://github.com/choltz95/sparse-precision-matrix-estimation/tree/master

**Relation To Broader Scientific Literature:**

To the best of my knowledge, this is the first paper introducing causality to study the world model in a controlled environment. If the authors can demonstrate that causality is necessary, I believe it will inspire follow-up research and have a significant impact in this area.

**Theoretical Claims:**

No theoretical claims

---

> ### Author Rebuttal · Authors · 2025-04-01
>
> We sincerely thank you for your thorough review and for the clear and detailed suggestions. We believe these suggestions further highlight the significance of the causality-based approach presented in the paper compared to correlation-based approaches.
>
> # Answer to questions
> ## Re Question 1 and 2
>
> Firstly, under the assumption made in the paper that hidden confounders may be present, a correlation-based method may not be applicable. The method we presented in the paper employs a causal discovery algorithm to extract a graph, accounting for the possible presence of latent confounders, from the attention matrix.
>
> As described below and empirically demonstrated, accuracy in generating legal tokens is more strongly associated with the causality-based approach presented in the paper than to the correlation-based methods, as measured by the confidence score (Section 4.3).
>
> You suggested 2 correlation-based methods to derive an adjacency matrix. The first is thresholding the attention matrix, and the second is thresholding the precision matrix. The first method is related to testing marginal independence (empty conditioning set), which was examined in Section 5.2.1, Figure 3a. In the second method the precision matrix represents pair-wise independence conditioned on all other nodes (only the largest possible conditioning set size, $n-2$). In relation to these methods, constraint-based causal discovery tests independence conditioned on a range of conditioning set sizes from 0 to the maximal possible, as needed. In Section 5.2.1 Figure 3 we demonstrate that statistical significance results obtained using empty conditioning sets are different than those obtained using size 1 (Figure 3a-3c). The aggregation of results from CI tests required by causal discovery improves the number of statistically significant cases (Figure 3d) differentiating legal from illegal move generation.
> Furthermore, given a graph $\mathbf{G}$, nodes' values are $\boldsymbol{X}=\boldsymbol{X}\mathbf{G}+\boldsymbol{U} \implies  \boldsymbol{X}=(\mathbf{I}-\mathbf{G})^{-1}\boldsymbol{U}$ (Equation 4). Treating attention as $\mathbf{G}$, instead of $(\mathbf{I}-\mathbf{G})^{-1}$ imposes restrictions on the graph.
>
> ### Empirical Evaluation
> For each game length we calculated confidence (Section 4.3). As in Figure 5, we binned confidence values and calculated mean accuracy. Results for Chess are in the tables below.
> It is evident that, except for a few cases, for both correlation-based methods there is no clear trend of accuracy with respect to confidence, as clearly evident for causality-based approach (Figure 5).
>
> **Attention thresholding**
> |Len|Bin 1|Bin 2|Bin 3|Bin 4|Bin 5|Bin 6|Bin 7|
> |-|-|-|-|-|-|-|-|
> |10|0.994|0.992|0.987|0.993|0.996|0.933|0.933|
> |15|0.998|0.987|0.977|0.967|0.978|0.988|1.000|
> |20|0.982|0.974|0.975|0.978|0.979|0.967|0.973|
> |25|0.889|0.960|0.964|0.957|0.969|0.986|1.000|
> |30|0.867|0.954|0.942|0.950|0.953|0.950|0.963|
> |40|0.667|0.980|0.970|0.943|0.938|0.928|0.938|
>
> **Precision-matrix thresholding**
> |Len|Bin 1|Bin 2|Bin 3|Bin 4|Bin 5|Bin 6|Bin 7|
> |-|-|-|-|-|-|-|-|
> |10|1.000|0.988|0.988|0.989|0.994|0.990|0.988|
> |15|1.000|0.982|0.977|0.980|0.982|0.979|0.975|
> |20|0.961|0.967|0.971|0.967|0.973|0.969|0.963|
> |25|0.941|0.956|0.964|0.963|0.966|0.967|0.980|
> |30|0.961|0.953|0.949|0.963|0.967|0.920|0.961|
> |40|0.945|0.940|0.936|0.934|0.925|0.882|0.933|
>
> ## Re Question 3
> A graph constructed using the last layer attention represents a graph over the output tokens.
> The last layer's attention output represents tokens (pre-training loss compares output tokens to input tokens), up to an MLP non-linear mapping, where MLP has no effect on inter-token relations and therefore does not influence causal structure learning. Earlier layers represent context (such as exogenous nodes in the SCM) for their following layer. See also the explanation following Equation 8. For example, in Othello intermediate layers may represent the board (as exhaustively tested and found by Li et al., 2023) which serve as context for the causal graph over game moves represented by the last layer. Furthermore, recent studies empirically show that the last layers have more weight in determining the next token, specifically in Othello GPT [1] and in LLMs in general [2,3,4].
>
> [1] How GPT learns layer by layer. Du et al., 2025.
> [2] Adaptive Large Language Models by Layerwise Attention Shortcuts. Verma et al., 2024.
> [3] The Mechanics of Conceptual Interpretation in GPT Models: Interpretative Insights. Aljaafari et al., 2024.
> [4] Exploring Concept Depth: How Large Language Models Acquire Knowledge at Different Layers? Jin et al., 2024.
>
> # Re Other comments or suggestions:
> * In Eq 9 both RHS and LHS are matrices. A hat over index $i$ represents the omission of the $i$-th row/column, as implicitly mentioned in the sentence before the equation. We will explicitly describe this notation.
> * We will clearly define the computation of the entropy values $H_\textrm{ind}$ and $H_\textrm{dep}$.

---

> > ### Comment · Reviewer_g9Sr · 2025-04-02
> >
> > Thanks for your detailed rebuttal. My concerns have been addressed and I have raised the score.

---

### Official Review · Reviewer_B5pU · 2025-03-11

**Overall Recommendation:** 3

**Summary:**

N/A

**Claims And Evidence:**

N/A

**Essential References Not Discussed:**

N/A

**Experimental Designs Or Analyses:**

N/A

**Methods And Evaluation Criteria:**

N/A

**Other Comments Or Suggestions:**

N/A

**Other Strengths And Weaknesses:**

N/A

**Questions For Authors:**

While I acknowledge my limited expertise in this domain and therefore express low confidence in this review, I have several concerns that lead me to recommend rejection. My concerns are as follows:

- First, I struggle to identify the core contribution of this paper. The authors explain the relationship between GPT and causal learning, but this connection seems rather obvious. GPT's autoregressive generation is inherently causal, and its architectural design enables efficient generation through causal mechanisms. The fact that GPT can learn causal knowledge about the real world is well-established. The attention mechanism learns token similarities through its similarity matrix, from which causal structures can naturally be extracted. None of this is surprising.

- Second, the paper's treatment of "world model" lacks precision. While this term is currently popular across various domains including robotics, autonomous driving, video generation, and 3D generation, it lacks a clear definition.

- Third, the practical implications of these findings are unclear. Understanding the relationships between GPT, causal learning, and world models, but how does this knowledge advance the field? Can it guide the development of better GPT models for real-world applications? The paper lacks substantial discussion of these practical considerations.

Due to my limited knowledge in this field, my review may be biased, and I welcome corrections from the authors and other reviewers.

**Relation To Broader Scientific Literature:**

N/A

**Theoretical Claims:**

N/A

---

> ### Author Rebuttal · Authors · 2025-04-01
>
> We sincerely thank you for your review and your perspective on the paper. We value your feedback and believe the following answers your concerns.
>
> * Re first point. We would like to clarify that autoregressive generation is not inherently causal. Often the attention in GPT is called 'causal' but it only means that a token is a function of previous tokens in the sequence. That is, the upper triangular part of the attention matrix is masked (zero). However, this is an oversimplification of the broader term "causal". In the paper we show that an attention value $\mathbf{A}_{i,j}$ describes the sum of all directed paths from node $j$ to node $i$, that is the effect node $j$ has on node $i$ accounting for all directed paths. Another difference between the attention and causal graphs is the set of encoded conditional independence relations. For example, consider the causal relations $X$ causes $Z$ and $Y$ causes $Z$, and no other relations: a graph: $X$ --> $Z$ <-- $Y$. The graph entails that $X$ and $Y$ are marginally independent, but dependent conditioned on $Z$. However, note that several causal graphs may entail the exact same independence relations (e.g., a hidden confounders may be present between $X$ & $Z$ and $Y$ & $Z$ instead of causal relations). The attention matrix only represents the correlation between $X,Y$ and $Z$ (the level of 'attention' $Z$ gives to $X$ and $Y$) and does not represent causality. The causal interpretation in this paper demonstrates that a causal graph, including hidden confounders, could be extracted, where the attention matrix represents the total effect a node has on another through all directed causal paths. That is, the $(i,j)$ element in the attention matrix $\mathbf{A}$ represents the sum of all the directed paths from node $j$ to node $i$ in the causal graph. We also introduce a novel confidence score for the learned graph using entropy of p-values of statistical tests used during the causal structure learning.
> * Re second point. We will clarify in the paper that a 'causal world model' describes the causal mechanism underlying the observations as well as probability distribution of hidden variables, as often used in the causal inference literature. Specifically, we will note that it is assumed that underlying each input sequence there exists a corresponding structural causal model (SCM, Section 3.2). That is, the causal world model consists of the causal mechanism that generates tokens.
> * Re third point. We discuss implications of the findings in Section 6 (Conclusions) and in Section 'Impact statement" (after Section 6. Conclusions; part of ICML 2025 template which does not count towards page limit). The implications include a) the ability for zero-shot causal discovery which can be beneficial in various scientific domains (e.g., understanding the mechanism by which a GPT trained on the protein space generates novel protein sequences [1]), b) measuring uncertainty of attention heads per input sequence, c) calibration during training (accuracy vs. causal confidence), and d) better human supervision by examining the causal mechanism by which a token is generated.
>
> [1] Ferruz, N., Schmidt, S., & Höcker, B. (2022). ProtGPT2 is a deep unsupervised language model for protein design. Nature communications, 13(1), 4348.

---

> > ### Comment · Reviewer_B5pU · 2025-04-02
> >
> > Thank you for the author's detailed response! My concern has been resolved, and I will raise the score.

---

### Official Review · Reviewer_v1kX · 2025-03-12

**Overall Recommendation:** 3

**Summary:**

The paper investigates whether a GPT trained for next-token prediction implicitly learns a causal world model, using an interpretation of the attention matrix as encoding a linear Gaussian SCM, first proposed in Rohekar et al. (2024). The introduce a causal discovery method for learning partially oriented causal graphs from the attention matrix using conditional independence testing. To indicate that the network is learning to represent a causal graph for a given sequence, they introduce a `structural confidence score’ R(A) which is the entropy difference between the conditional independence test p-values for detected dependencies and independencies. Experimental results show that sequences with higher structural confidence scores correlate with correct legal move predictions, and that pruning low-confidence attention heads does not affect performance, whereas pruning high-confidence heads does.


## update after rebuttal

Following the authors rebuttal, I have updated my score and am leaning towards acceptance, but with a low confidence due to my own lack of familiarity with the mechanistic interpretability aspects of the paper and how faithfully they map over to the causal claims.

**Claims And Evidence:**

The main claims are supported by correlating high R(A) with legal move accuracy. However, it is not immediately clear to me why this implies the causal interpretation is valid, when this correlation could have other explanations. It is also not immediately clear to me why this was the experiment they chose to validate the interpretation, rather than the more obvious ones of comparing the learned causal structure to the true causal structure, or making interventions and seeing how the learned causal structure changes (see questions below).

Overall, I find the proposal quite compelling but am confused by the way they have gone about validating it.

**Essential References Not Discussed:**

There are some papers the authors could discuss below, that motivate for the papers results (i.e. why should we even expect the network to have learned causal structure), [1] showing that this is required for out of distribution generalization, and [2] giving examples where causal world models are detected using mechanistic interpretability. These are not essential, but could strengthen the argument.

[1] Robust agents learn causal world models (Richens et. al).
[2]Transformers Use Causal World Models in Maze-Solving Tasks (Spies et. al.)

**Experimental Designs Or Analyses:**

I have not assessed these in depth, but they appear to be rigorous, confidence intervals are given, etc.

**Methods And Evaluation Criteria:**

The choice of domains is reasonable and has been studied before in this context

**Other Comments Or Suggestions:**

There are lots of grammar errors in the paper, which made some key parts of it hard to follow.
Abstract “Are generative pre-trained transformer (GPT) models, trained only to predict the next token, implicitly learn a world model from which a sequence is generated one token at a time?”, suggest replacing “Are” with “Do”. “suggesting a causal world model that arises from this interpretation.” this sentence doesnt make sense to me,that the causal world model arises from your interpretation?
“Note that even if some of the nodes are latent confounders is still (I − G) −1 triangular”
Many more, would recommend some polishing

**Other Strengths And Weaknesses:**

I find the approach compelling and the overall quality of the paper good, but it is perhaps somewhat incremental in extending Rohekar et al. (2024) to GPT models?

**Questions For Authors:**

1. Can you provide a rigorous justification for the claim that D^-1 A \simeq (I-G)^-1 ? Have you tested this on synthetic data with a known causal graph, and found that this correspondence approximately holds?
2. Im confused why you don’t include an interventional study, where you change the game rules (e.g., modifying Othello transition dynamics) to verify if the causal structure you recover using your interpretation changes according to this intervention. This feels like much more direct evidence for the causal interpretation than correlating the models ability to predict legal moves for a given sequence with the certainty in the causal structure returned by your causal discovery algorithm.
3. “in cases where the confidence in recovering structures from the attention matrices is low, GPT generally fails to generate a token that adheres to the game rules”. This doesnt obviously imply that “GPT implicitly learns to represent causal structures in attention heads”. Can you explain this connection more?
4. How does the learned causal structure compare to the actual game rules? Could you provide qualitative examples of inferred causal graphs? It is not clear from R(A), which measures only how concentrated the p-values are for the conditional independence tests, that the model is learning the correct causal graph. Are there other reasons that a high R(A) could correlate with legal moves? E.g. move legality is determined by a few most recent states, resulting in a high localised attention pattern, which could perhaps show up as a high R(A)?
5. Given that multiple interpretations of attention heads exist (e.g., memory retrieval, information diffusion), what makes the causal interpretation the most compelling? (note [1] could give some justification). It could be beneficial to include in the paper a discussion of other interpretations, and how these could explain the observed correlation between R(A) and performance? (e.g. consider in your analysis alternative hypotheses beyond the causal structure learning hypothesis you are proposing).

**Relation To Broader Scientific Literature:**

The work aligns with recent findings that transformers learn implicit world models (Li et al., 2023; Nanda et al., 2023) and extends these ideas by applying an adapted version of Rohekar et al. (2024) for GPT models. The paper does not cite alternative interpretations of attention (e.g., information flow, memory retrieval, statistical smoothing), and it may be worth discussing how the causal interpretation of attention relates to these other interpretations.

**Theoretical Claims:**

The key theoretical assumption, that D^-1 A \simeq (I-G)^-1, is plausible but lacks rigorous derivation beyond conceptual arguments. It would be beneficial to validate this claim on synthetic data with a known causal structure.

---

> ### Author Rebuttal · Authors · 2025-04-01
>
> Thank you for your detailed feedback, insights, and important questions. Addressing your review improves the over clarity of the paper and emphasizes the significance of the contribution.
>
> # Re Questions for Authors
>
> 1. The relation $\mathbf{D}^{-1}\mathbf{A} = (\mathbf{I}-\mathbf{G})^{-1}$ is not an assumption, but rather a result by considering each token as a node in an SCM (the sentence just before equation 8). Since attention calculates $\mathbf{Z}=\mathbf{A}\mathbf{V}$ (Equation 1) and the SCM calculates $\boldsymbol{X}=(\mathbf{I}-\mathbf{G})^{-1}\boldsymbol{U}$ (Equation 4), we equate the outputs: token embedding and SCM node value. This is described two sentences before Equation 9, but we will clarify this point more explicitly in the paper.
>
> 2. A ground truth of causal graphs is not available for most domains. In the strategy games of Othello and Chess, since there are multiple next legal moves, a causal graph over a given set of game moves may contain information about the strategy of the player (in case of a real game) in addition to the game rules. Moreover, it is unclear how to modify the game rules such that a sequence of game moves is legal for both the original rules and the modified rules, while making sure the set of next legal moves entailed by the modified rules does not overlap with that entailed by the original moves. That is, for a fair comparison the input sequence should be legal for both the original and intervened rules, and for correctly testing, the next legal move should be different. Nevertheless, at the request of Reviewer g9Sr, we evaluated the accuracy of legal move generation as the function of confidence score (Section 4.3) calculated for correlation-based methods they suggested. See also Figure 3a vs. Figure 3d in the ablation study (Section 5.2).
> We believe your concern is additionally addressed in our reply to their Questions 1 and 2. We did not find an increase in legal move generation accuracy as a function of the confidence score for correlation-based methods (as found for the learned causal graph).
>
> 3. For a given sequence of tokens and a legal next token generation, there may be multiple, equally minimizing the loss, attention matrices. During pre-training, these attention matrices were not restricted to have high confidence for determining high-order conditional independence (CI) relations between tokens (in comparison to marginal pair-wise correlations). We examined confidence of attention matrices generating legal vs illegal token generation. In all our experiments we found a correlation between the causal confidence score, which was calculated during causal structure learning, and the accuracy of generating legal moves. Confidence decreases when uncertainty increases in CI tests decisions used for constructing the causal graph structure (e.g., an edge is removed if conditional independence is found, and edge orientation is determined based on the conditioning set that disconnected two nodes). We will add a corresponding clarification this in the paper.
>
> 4. We provided a qualitative examples of causal graphs learned for the first few moves in Othello and Chess games in Figure 1 and Figure 6, respectively. These graphs are actual results. After these few moves it becomes challenging to follow the true causal relations. For example, during the Othello game, previously placed pieces may flip their color multiple times affecting the set of legal moves. Next, we regard the possibility of having high $R(\mathbf{A})$ due to the last few moves effecting the next move. In our Ablation study in Figure 3 we compare (a) marginal (in)dependence relations to (d) conditional independence relations that are required to construct causal graph. It is demonstrated that the difference in $R(\mathbf{A})$ between legal and illegal token generation is more prominent when considering independence relations required for constructing a causal graph. As mentioned in our reply to your Question 2, we also examined the experiment in Figure 5 for correlation-based methods.
>
> 5. The main difference between the causal interpretation and other interpretation is the incorporation of conditional independence relations having various conditioning set sizes rather that treating the attention values as weights. As mentioned in our answer in Question 1, by relating output embeddings computed using the attention matrix to the node values in an SCM we obtain the relation  in Equation 9. By equating the covariance matrices (Equation 8) we can employ a constrain-based causal discovery algorithm. In our experiments we demonstrated that a confidence score computed from conditional independence relations provides a better differentiation between legal and illegal token generation compared to marginal independence relations.
>
> Finally, we found the papers you suggested to include the the introduction to strengthen the significance of this paper's results. As suggested, we will add a related discussion.

---

> > ### Comment · Reviewer_v1kX · 2025-04-02
> >
> > Thank you for your detailed responses. I have updated my scores.
> >
> > The authors have written a paper at the intersection of these two fields, which is an important and under explored area, but makes it challenging to get good reviews. But it would be a shame if this factor prevented publication and in doing so further exploration of this intersection. So while my score is a weak accept, reflecting my uncertainty in the validity of the papers claims, I think the paper should probably be accepted.

---

### Official Review · Reviewer_EPcN · 2025-03-13

**Overall Recommendation:** 3

**Summary:**

The work explores whether or not GPT style models learn a causal world model implicitly without explicitly being trained to do so by using the predict the next token objective. This is done in Othello and chess, and the theoretical formalization paired with the empirical results strongly suggest that GPT style models do learn a causal world model. The strong results in Figures 4, 5 support this, and they even show this extends to out of distribution outputs (Figure 7).

**Claims And Evidence:**

The claims in the submission are supported by strong and clear evidence. The authors support their claim regarding causality both with a theoretical formalization and then empirical results.

**Essential References Not Discussed:**

N/A

**Experimental Designs Or Analyses:**

I thoroughly checked the validity/soundness of all experiments regarding Figures 3-5, 7-8. I did not see any issues and believe the experimental design strongly support the claims.

**Methods And Evaluation Criteria:**

The benchmark datasets, consisting of Othello and Chess, make sense for the problem of determining whether world models learn a causal structure.

**Other Comments Or Suggestions:**

- Figure out a better way other than n grams to show the test and train distributions are different
- Do more experiments regarding the extent to which OOD generalization occurs
- The figures were not very clear to me, the captions could have better encapsulated what was being done.
- It would have been nice to see non game benchmarks, i.e. math benchmarks or some other problems with causal structure, but the benchmarks were sufficient for the claims.

**Other Strengths And Weaknesses:**

## Strengths
- Very important work to formalize theoretically that GPTs can learn a causal world model
- strong theoretical analysis of causality in attention and generally strong results (Figures 4, 5) to support them

## Weaknesses
- It's still unclear the extent to which generalization occurs, even though we know it worked for the OOD test set
- Although in Figure 3 the results are statistically significant, they were not statistically significant very often, especially since the p value was not very high at 0.05%.
- It's unclear whether these results would hold for language models not trained solely on chess/othello data, or data that is noisy/imperfect

**Questions For Authors:**

- The paper states "determined threshold of significance level (α). It is important to note that there is a one-to-one correspondence between the results of these CI test and the entailed causal structure. That is, a causal structure can be represented uniquely by a set of CI tests and their results."
	- How? This needs a citation, or is there a mistake?
- Why is positional encoding not used?
	- Do they mean positional encoding specifically for the board?

**Relation To Broader Scientific Literature:**

There are two main ways this is related to the broad scientific literature. First, is the relation to world models and autoregressive (GPT) style models. Second, is the relation to causality/causal inference. The results bring link two domains together, and effectively demonstrate that GPT style autoregressive models can learn a world model which understands causal factors at play and can reason about what this causality implies.

**Theoretical Claims:**

I briefly checked through all of the proofs for theoretical claims and did not find any issues

---

> ### Author Rebuttal · Authors · 2025-04-01
>
> Thank you for the detailed review and suggestions for improvement. Your suggestions will improve the clarity and emphasize the significance of the proposed approach and findings presented in the paper.
> We also thank you for the many suggestions for future work.
>
> **Re Other Comments or Suggestions**
>
> Upon your interesting suggestions, in our future work we will extend the analysis for OOD data in different domains. Specifically, we plan to explore improvements for pre-training relying on the findings in the paper. For example, we plan to explore the effect of OOD generalization after calibrating token-generation accuracy with respect to causal confidence (Equation 10).
> Finally, we will improve the clarity of figures' captions as requested and correct grammatical errors and typos.
>
> **Re Questions for Authors**
>
> * The full set of CI tests having all possible conditioning set sizes has a one-to-one correspondence with a causal graph (up to Markov equivalence). A causal-discovery algorithm uses a sub-set of this exhaustive set of CI tests. A sound and complete algorithm, like the one we use in the paper, creates a one-to-one correspondence between this subset and a Markov equivalence class. We will add this clarification along with a citation (Spirtes et al., 2009).
>
> * Yes. Positional encoding specifically for the board was not used (in accordance with the work of Li et al., 2023). Positional encoding was (also) not used by Li et al., 2023 who trained a GPT without domain information, such as the presence of a board and its 2D alignment on which the game moves take place. We will clarify this.

---

> > ### Comment · Reviewer_EPcN · 2025-04-07
> >
> > [Accidentally posted as official comment]
> > Thanks for the rebuttal and clarification. It would have been great to see some of the weaknesses/comments addressed regarding whether or not these results hold for noisy data as well as data that is OOD and has been generated/curated/validated as being different than the training set in other ways (i.e. more results on this). Additionally, the concerns regarding whether or not language models not trained solely on chess/othello data or noisy data would still learn such causality is unclear, limiting the application of this work. This is especially important as recent work suggests that generalization/reasoning is very much tied to data quality [1], which could be the same for causality.
> >
> > Additionally, my general lack of understanding of the literature on causality makes me less confident in my initial score.
> >
> > Thus, I am reducing my score slightly to a 3. The general response to reviewers was strong, although further addressing my points with empirical evidence would be helpful.
> >
> > [1] https://arxiv.org/pdf/2503.07604

---

> > > ### Author Response · Authors · 2025-04-09
> > >
> > > Thank you for your follow up questions and thank you for your initial positive feedback. We apologize for this late response, as we received your follow up questions just yesterday. We believe our answers can help address your concerns and clarify points you raised.
> > > Upon your suggestion, we further created a noisy data and examined the method we presented in the paper. Specifically, we added noise to each test sequence of game moves by replacing $p_{noise}$ percentage of moves with random illegal moves (recall that the initial test sequence was sampled from the set of all possible legal moves, oblivious to winning the game). This process constitutes OOD test sequences as these kind of illegal sequences are inherently different from the training sequences (recall that GPT was pre-trained on real games played with the intention of winning).
> > >
> > > The following tables, summarize the results for the Othello game for different game lengths and for different noise levels ($p_{noise}$). We split the test sequences into those that resulted in GPT generation of a) illegal next token and b) legal next-token. We calculated the mean causal confidence (Equation 10) for each group and report their difference.
> > >
> > > First, the accuracy of the model in generating legal next tokens for different noise levels is provided in the following table.
> > >
> > > |$p_{noise}$|17 moves|20 moves|22 moves|25 moves|30 moves|
> > > |:-----:|:------:|:------:|:------:|:------:|:------:|
> > > | 0.00    | 0.892 | 0.906 | 0.909 | 0.920  | 0.936 |
> > > | 0.05 | 0.858 | 0.835 | 0.841 | 0.842 | 0.859 |
> > > | 0.10  | 0.794 | 0.779 | 0.784 | 0.767 | 0.776 |
> > > | 0.15 | 0.729 | 0.736 | 0.744 | 0.713 | 0.699 |
> > > | 0.20  | 0.663 | 0.702 | 0.673 | 0.667 | 0.664 |
> > > | 0.30  | 0.580  | 0.584 | 0.616 | 0.599 | 0.579 |
> > >
> > > Next, the difference between the causal confidence of legal token generation and illegal token generation is provided in the following table. Positive values indicate that the mean confidence of legal token generation is higher than that of illegal token generation (higher is better).
> > >
> > > |$p_{noise}$|17 moves|20 moves|22 moves|25 moves|30 moves|
> > > |:-----:|:------:|:------:|:------:|:------:|:------:|
> > > | 0.00     | 0.091  | 0.226  | 0.175  | 0.102  | 0.220   |
> > > | 0.05  | 0.081  | 0.189  | 0.137  | 0.139  | 0.148  |
> > > | 0.10   | 0.089  | 0.193  | 0.151  | 0.153  | 0.178  |
> > > | 0.15  | 0.176  | 0.212  | 0.143  | 0.147  | 0.095  |
> > > | 0.20  | 0.178  | 0.170   | 0.148  | 0.119  | 0.154  |
> > > | 0.30   | 0.037  | 0.143  | 0.034 | 0.025  | 0.015  |
> > >
> > >
> > > It is evident that the causal confidence of legal token generation is consistently higher than that of illegal token generation for all the test noise levels (positive values). This extends the conclusion in the paper to noisy and OOD test data.
> > >
> > > We would like to note that training data does not take part in the causal discovery process. In this paper we assume that the GPT pre-training process trains the attention mechanism to capture relations between tokens in the input sequence. For applications in which noise is expected in the data, this method requires the attention to be able to capture correct independence relations as described in the paper. As long as the trained GPT can faithfully represent relations between tokens through attention, the causal discovery part will be free of errors. Note that there is no training involved for the causal discovery part. In general, causal discovery under unknown noise in highly non-linear relations is an unsolved problem. The use of large datasets by GPT to convert these noisy and non-linear relations to stable linear relations (via self-supervision) for causal discovery is one of this paper's contributions (section 4.2). We will clarify this in the paper.
> > >
> > > Finally we would like to mention a few potential applications beyond Othello and Chess that can readily benefit from this paper's contribution. These include protein sequence generation [1, 2] and material design [3]. In protein sequence generation, tokens are amino acids and in material design tokens describe the atomic structure (e.g., via SMILES format). For example, domain experts may utilize existing foundation models for these domains to reason about causal relations between amino acids or between molecules.
> > >
> > > 1. Ferruz, Noelia, Steffen Schmidt, and Birte Höcker. "ProtGPT2 is a deep unsupervised language model for protein design." Nature communications 13.1 (2022): 4348.
> > > 2. Rives, Alexander, et al. "Biological structure and function emerge from scaling unsupervised learning to 250 million protein sequences." Proceedings of the National Academy of Sciences 118.15 (2021): e2016239118.
> > > 3. Soares, Eduardo, et al. "A large encoder-decoder family of foundation models for chemical language." arXiv preprint arXiv:2407.20267 (2024).

---

### Decision · Program_Chairs · 2025-05-01

**Decision:**

Accept (poster)

**Comment:**

This paper investigates an important and interesting research question: Can GPT-style models trained only to predict the next token implicitly learn a world model? The authors provide an interpretation of the attention mechanism in GPT from a causal perspective to validate this issue. In addition, the experimental results are strong, which can support the authors' claim.

In the rebuttal period, the authors have made substantial efforts to address the reviewers' concerns and provided a detailed response, including additional experimental results to further verify the effectiveness of the proposed approach, as well as a clear explanation of concepts, definitions, and empirical findings. Most of the major concerns raised by the reviewers have been addressed.

Given the positive feedback from all reviewers and the improvements made, I recommend accepting the paper for publication.